# Study of Force-Frequency Characteristics in AT-Cut Strip Quartz Crystal Resonators with Different Rotation Angles

**DOI:** 10.3390/s23062996

**Published:** 2023-03-10

**Authors:** Gang Yang, Xianhe Huang, Ke Tan, Qiao Chen, Wei Pan

**Affiliations:** School of Automation Engineering, University of Electronic Science and Technology of China, Chengdu 611731, China

**Keywords:** force-frequency characteristics, strip quartz crystal resonator (QCR), rotation angles, force sensor

## Abstract

This paper investigated the force-frequency characteristics of AT-cut strip quartz crystal resonator (QCR) employing finite element analysis methods and experiments. We used the finite element analysis software COMSOL Multiphysics to calculate the stress distribution and particle displacement of the QCR. Moreover, we analyzed the impact of these opposing forces on the frequency shift and strains of the QCR. Meanwhile, the resonant frequency shifts, conductance, and quality factor (Q value) of three AT-cut strip QCRs with rotation angles of 30°, 40°, and 50° under different force-applying positions were tested experimentally. The results showed that the frequency shifts of the QCRs were proportional to the magnitude of the force. The highest force sensitivity was QCR with a rotation angle of 30°, followed by 40°, and 50° was the lowest. And the distance of the force-applying position from the *X*-axis also affected the frequency shift, conductance, and Q value of the QCR. The results of this paper are instructive for understanding the force-frequency characteristics of strip QCRs with different rotation angles.

## 1. Introduction

The quartz crystal resonator (QCR) consists of a quartz substrate sandwiched between two electrodes. It is widely used as a sensor for measuring external parameters, such as pressure [1,2], mass [3,4,5], and humidity [6,7,8], due to its compact size, high frequency, stability, and good repeatability.

In 1947, Bottom’s discovery of the force-frequency characteristics established the linear relationship between the resonant frequency of a QCR and the magnitude of an external force [9]. This relationship was attributed to the modifications in the elastic model of the quartz crystal caused by the applied force, which led to changes in the propagation velocity of the resonant wave. Bottom’s finding served as the theoretical foundation for the development of quartz crystal force sensors. Murozaki et al. developed a small-sized load sensor that was highly sensitive using an AT-cut QCR. Through the use of this sensor, multiple biological signals, such as breath, heartbeat, and posture, can be detected with high stability [10]. Meanwhile, Asakura et al. have developed a compact force sensor based on AT-cut QCR, utilizing a simple structure and a novel retention mechanism for the QCR [11]. Additionally, Sakuma et al. created a cantilever-shaped QCR force sensor probe by building upon their previous work on force sensors. By utilizing this probe, high-resolution force sensing can be achieved [12]. In the past period, researchers have extensively studied the force-frequency characteristics of disc-type QCRs. Ratajski defined the force-frequency coefficients of disc-type QCRs through experimental studies [13]. Dauwalter’s research revealed that changes in temperature could have an impact on the force-frequency coefficient, with the extent of this effect being determined by the azimuth angle between the X crystallographic axis and the direction in which force is applied [14]. Building upon Ratajski’s earlier work, Ballato et al. investigated the force-frequency effect in doubly rotated quartz resonators [15]. Lee, in a separate study, used the theory of initial finite deformation superimposed on incremental elastic deformation to summarize the equations for calculating the frequency shift and established the two-dimensional control equations for the vibration of piezoelectric quartz plates by Mindlin’s general procedure of power-series expansions of displacements and body forces [16].

Further, Janiaud et al. presented a solution to the problem of stress bias effects in anisotropic circular plates [17]. Jing et al. studied the effect of electrodes on the force-frequency characteristics of QCRs [18]. Bao et al. reported the first experimental measurement of the stress-induced frequency shifts of thickness-shear modes in a rotated Y-cut QCR [19]. To reduce the temperature and other interference factors and improve force-sensitivity, Chen et al. designed a multi-electrode force-sensitive resonator cluster based on the stress distribution results whose force-frequency coefficient reached 10,167 Hz/N [20]. 

Subsequently, QCRs with shapes other than disc-type have also been investigated for their force-frequency characteristics. Wang et al. discussed the resonant frequency changes in the thickness-shear vibration of symmetrical incomplete circular AT-cut QCRs, which were used as sensing elements in digital force sensors or pressure transducers by the application of radial forces [21]. Recently, a mathematical finite element method(FEM) model was developed by Mohammadi et al. to study the frequency change of a square AT-cut quartz crystal subjected to a pair of opposing forces on different points of its edge [22]. Those results indicate that the frequency shift is proportional to the magnitude of the force when the force is applied to the side of QCRs of different shapes.

In addition to the QCR shapes mentioned above, more and more literature is focusing on the theory and applications of strip QCRs. It is worth noting that strip QCRs can be customized in terms of shape, size, and patterns to suit different sensing applications. This flexibility allows for more precise and sensitive measurements. Lee et al. studied the vibrations of doubly-rotated strip quartz driven by the lateral electric field applied to a pair of electrode-plated and traction-fresh edges [23]. Later, Lee and Wang derived one-dimensional equations of motion for AT-cut strip QCRs based on the two-dimensional equations of motion without considering the piezoelectric effect [24]. On the basis of this, Wang et al. extended the one-dimensional equations to incorporate the piezoelectric effect, resulting in a more complex set of coupled equations that were considerably larger than the ones used for analyzing mechanical vibrations. And the forcing frequency on the resonance frequency, capacitance ratios, and vibration mode shapes were obtained [25]. Apart from these, Nosek et al. computed the resonance temperature coefficients of the small Y-cut quartz strip resonators in different rotation angles [26]. Additionally, rectangular QCRs have been used as strain sensors for measuring mechanical quantities in Grossmann’s research [27]. Then, the acceleration sensitivity performance of AT-cut strip quartz crystal oscillators was improved by Fry et al. [28]. Next, Zhao et al. studied the thickness-shear vibrations of an x-strip monolithic piezoelectric plate made from AT-cut quartz crystals with two unequal electrode pairs [29]. 

While numerous studies have examined the behavior of AT-cut strip QCR under various conditions, there is still a lack of comprehensive investigation into certain critical parameters. Specifically, little attention has been given to exploring the frequency shift and stability of a strip QCR when subjected to opposing forces at different positions along its sides. This is a crucial area of research that requires further investigation, as it has significant implications for the performance and stability of strip QCRs in a range of practical applications.

In this paper, we first analyzed the particle displacement and stress distribution of the AT-cut strip QCR under forces in the direction of the *X*-axis using the finite element analysis software COMSOL Multiphysics. And then, the frequency shift of the QCR subjected to a pair of opposing forces under different force-applying positions was also simulated. Furthermore, we also studied the variation of the strains of the QCR with the rotation angles. Finally, the experiment was conducted for three sets of AT-cut strip QCRs with different rotation angles. The resonant frequencies, conductance, and quality factor (Q value) of the QCRs when a pair of opposing forces was applied at different positions were also compared and analyzed.

## 2. Theory

Figure 1 is a schematic diagram of a disc-type QCR subjected to two opposing radial forces. For a QCR with a defined angle of cut, the frequency shift generated by the external force is related to several parameters. The Ratajski force-frequency coefficient of a disc-type AT-cut quartz crystal resonator subjected to radial forces is defined as [13]:(1)$$K_{f}(\psi ,\theta )=\frac{\Delta f}{f}\times \frac{1}{F}\times \frac{nD}{f}$$
where F is the value of the applied force, f is the resonant frequency of the QCR, Δf is the corresponding frequency shift caused by force, D is the diameter of the crystal disk, and n is the number of overtones. It can be seen that the force-frequency coefficient is a function of the cut angle, θ, and the azimuth, ψ, of the applied force. The azimuth, ψ, is marked in Figure 1, which is the angle between the *X*-axis of the QCR and the direction of the applied force.

$K_{f}$ is normalized to the geometry of QCR, resonant frequency, and the number of overtones, which facilitates comparison of the force-frequency characteristics of different resonators. However, for studying the characteristics of QCR, the force sensitivity coefficient, $S_{F}$, is more convenient and straightforward. When a QCR with resonant frequency (f) is subjected to a radial force (F), its resonant frequency becomes $f_{0}+\Delta f$, and the force sensitivity coefficient is defined as [20]:(2)$$S_{F}=\frac{\Delta f}{F},K_{f}=S_{F}\cdot \frac{n\cdot D}{f^{2}}$$

The unit of the Ratajski coefficient, $K_{f}$, is m·s/N, and the force sensitivity coefficient, $S_{F}$, is Hz/N.

The force-frequency effect of QCR can be considered an elastic dynamic problem of superimposing a small amplitude elastic vibration based on the initial finite elastic deformation caused by an external force. The relative frequency shift of QCR in the thickness-shear vibration mode under the initial stress can be expressed as [16]:(3)$$\frac{\Delta f}{f_{0}}=U_{1,1}^{\left(0\right)}+\frac{1}{2C_{66}}\left[C_{661}E_{1}{}^{\left(0\right)}+C_{662}E_{2}{}^{\left(0\right)}+C_{663}E_{3}{}^{\left(0\right)}+C_{664}E_{4}{}^{\left(0\right)}\right]$$
where $f_{0}$ is the resonant frequency without stress, Δf is the frequency shift caused by the stress, and $E_{i}^{\left(0\right)}$ is the zero-order strain component. $C_{66}$ and $C_{66i}$ are the second-order and third-order stiffness coefficients of the quartz crystal, respectively. Through this equation, the frequency shift of the QCR can be calculated directly from its strain. 

When the force is exerted in the strip QCR, the analysis can be done with Mindlin plate theory, the basic assumption is that the plate thickness is much smaller than the length and width of the plate [30]. The most important application of the theory is the high frequency vibrations of crystal plates [31]. By neglecting the higher order Mindlin strain terms, zeroth-order strains in Equation (3) can be obtained by [22]:(4)$$\begin{array}{l} E_{1}^{\left(0\right)}=S_{11}T_{11}+S_{13}T_{33}+S_{15}T_{13}\\ E_{2}^{\left(0\right)}=S_{21}T_{11}+S_{23}T_{33}+S_{25}T_{13}\\ E_{3}^{\left(0\right)}=S_{31}T_{11}+S_{33}T_{33}+S_{35}T_{13}\\ E_{4}^{\left(0\right)}=S_{41}T_{11}+S_{43}T_{33}+S_{45}T_{13} \end{array}$$
where $S_{ij}$ is the elastic constants of quartz crystal, $T_{ij}$ is the stress component of quartz crystal, and $E_{i}{}_{j}$ is the zeroth-order strain component of quartz crystal.

## 3. Simulation

The finite element method has been applied in many scientific computing fields due to its high calculation accuracy and wide application range [32,33,34]. In this paper, we used the finite element analysis software COMSOL Multiphysics to analyze the frequency shifts of the QCR under different rotation angles and different force-applying positions. As shown in Figure 2a, a FEM model of AT-cut strip QCR with a length of 9 mm, a width of 2.5 mm, and a thickness of 0.277 mm was created. The electrode material was selected as silver, with a theoretical value of a resonant frequency of 6 MHz. When an alternating voltage was applied to it through the electrode, mechanical oscillation was generated due to the piezoelectric effect of the quartz crystal. As shown in Figure 2b, when the oscillation frequency caused by the applied voltage was equal to the resonant frequency of the QCR, a resonant state was reached. It can be seen that the energy of the QCR was mainly gathered inside the electrode, and there was an obvious thickness shear vibration in the *X*-axis direction at a resonant frequency of 6.0905 MHz.

Since quartz is a typically brittle material with weak properties in tension, it is usually evaluated using compression tests [35]. The particle displacement contours and stress distribution of the cross-section of the QCR with 0.5 N, 1 N, and 1.5 N forces applied along the *X*-axis, respectively, were simulated by COMSOL software, as shown in Figure 3, Figure 4 and Figure 5. By analyzing the simulation results, the following conclusions can be summarized:(1)The particle displacement and stress distribution at various points in the cross-section was non-uniform under the action of external forces. Near the location of the point where the force was applied, high-intensity stresses and displacements were generated, and the displacements and stresses achieved maximum values.(2)With the increase of the applied force, the displacement and stress distribution at various points showed different degrees of change. In particular, the displacement and stress showed a significant increasing trend near the location of the point where the force was applied.(3)The particle displacement contour plot and the normal stress distribution were symmetric about the *X* and *Z* axes, while the shear stress was antisymmetric about the *X* and *Z* axes.(4)The magnitude of the stress and displacement was basically proportional to the magnitude of the applied force.

**Figure 3 sensors-23-02996-f003:**
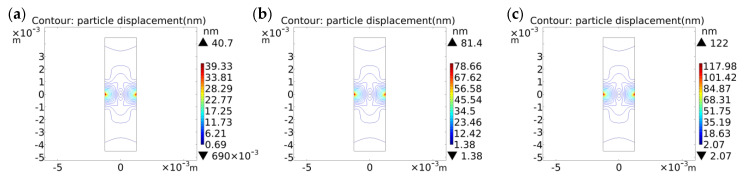
Particle displacement contour plot of the cross-section. (**a**) under the force of 0.5 N; (**b**) under the force of 1 N; (**c**) under the force of 1.5 N.

**Figure 4 sensors-23-02996-f004:**
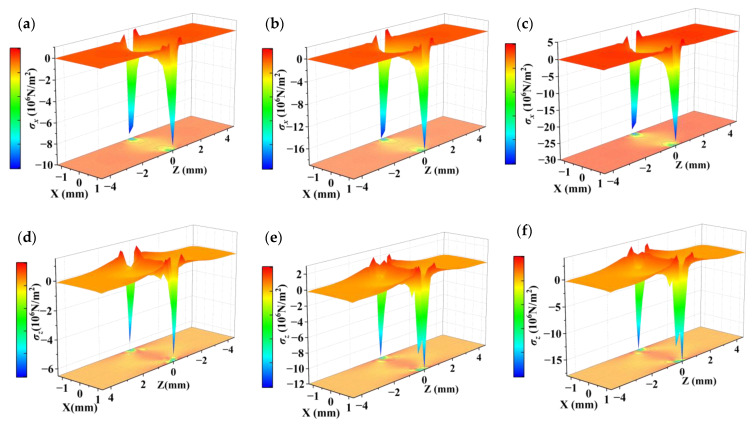
Distribution of normal stress in cross-section. (**a**) $\sigma_{x}$ under the force of 0.5 N; (**b**) $\sigma_{x}$ under the force of 1 N; (**c**) $\sigma_{x}$ under the force of 1.5 N; (**d**) $\sigma_{z}$ under the force of 0.5 N; (**e**) $\sigma_{z}$ under the force of 1 N; (**f**) $\sigma_{z}$ under the force of 1.5 N.

**Figure 5 sensors-23-02996-f005:**
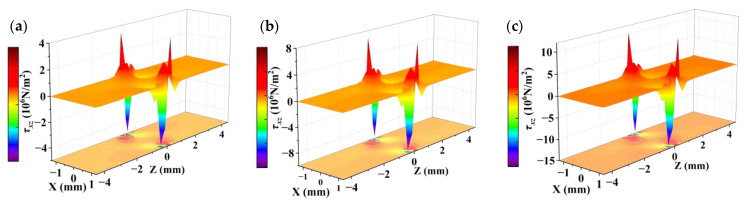
Distribution of shear stress $\tau_{xz}$ in cross-section. (**a**) under the force of 0.5 N; (**b**) under the force of 1 N; (**c**) under the force of 1.5 N.

From Equations (3) and (4), the frequency shift of a QCR under applied force was related to the stress variation within the wafer. This stress variation could lead to differences in the force-frequency characteristics of the QCR for electrodes located at different positions. To design a quartz crystal-based force sensor with high sensitivity, a finite element simulation analysis can be used to identify the locations where stress variation in the QCR is high. The electrodes can then be placed at these locations to optimize the sensitivity of the force sensor.

Compared to the disc-type QCR in Figure 1, which can only be applied radial forces in the X–Z plane at different angles, the strip-shaped QCR has more force-applying configurations, as shown in Figure 6. The configurations were created by rotating the strip QCR around its *Y*-axis and changing the force loading position, and the rotation angle increased from 0° to 180° in steps of 10°. At each rotation angle, the corresponding frequency shift was calculated by changing the distance of the force-applying position from the *X*-axis with a fixed force of 0.5 N. As shown in Figure 7, the curves were symmetric about the *X*-axis due to the monoclinic symmetry of the AT-cut quartz crystal.

Evidently, the frequency shift was maximal when the force was applied along the *X*-axis at a distance of 0 mm. This is because the wave propagated in the X-direction, and the QCR’s primary vibrational direction aligned with the direction of the applied force at this position. And it is worth noting that the frequency shift was approximately maximized at a rotation angle of around 20° when the force-applying position was situated at a distance of 0 mm from the *X*-axis rather than at 0° as might be expected. This phenomenon could be attributed to the anisotropic nature of the quartz crystal. Furthermore, when the QCR was rotated to an angle of 60°, the resonant frequency remained almost constant under an applied force. Therefore, for optimal sensitivity when designing strip force sensors, the rotation angle of the QCR should be around 20°. And these simulation results were consistent with findings reported by other researchers regarding disc-type AT-cut QCRs [13,20].

Upon Equation (3), the occurrence of zero frequency shifts in some rotation angles depends on the initial stress components and the corresponding zero-order Mindlin strains. Variation of the rotation angles results in the variation of these four strain components. By using Equation (3), a small strain component $E_{4}^{\left(0\right)}$ was neglected, and applying $E_{1}^{\left(0\right)}=U_{1,1}^{\left(0\right)}$ [16], one obtains:(5)$$\Delta f=f_{0}\left[(1+\frac{C_{661}}{2C_{66}})E_{1}^{\left(0\right)}+\frac{C_{662}}{2C_{66}}E_{2}^{\left(0\right)}+\frac{C_{663}}{2C_{66}}E_{3}^{\left(0\right)}\right]$$

Equation (5) describes how the zero order strain components and frequency shifts are related, which is shown that the frequency shift not only depends on the three strains, but also on their respective coefficients $\left(1+\frac{C_{661}}{2C_{66}}\right)$, $\frac{C_{662}}{2C_{66}}$, and $\frac{C_{663}}{2C_{66}}$. Figure 8 shows the variation of the strain of the QCR with the rotation angles. When the rotation angle was around 20°, the frequency shift was the largest because the sum of the products of the strain components and the coefficients was the highest. On the other hand, when the rotation angle was 60°, the stress component $E_{3}$ was nearly zero, and the strain components $E_{1}$ and $E_{2}$ satisfied the condition that Equation (5) was equal to zero. As a result, the frequency shift was approximately zero.

## 4. Experiment

We investigated the effect of different force-applying positions and forced magnitudes on the frequency shifts and stability of the AT-cut strip QCR through a force-frequency effect experiment at a room temperature of 25 °C. Figure 9 shows the experimental setup. It can be seen that the QCR was supported at the top and bottom by designed fixtures. The mass of the weights was applied to the side of the QCR using a tray and fixtures, and a force loading range of 0–3 N was achieved by adding standard weights. The force was applied to the two long sides of the strip QCR in the thickness direction while the holder held the two short sides in place. To investigate the effect of force-applying position on the behavior of the QCR, the center of the strip QCR could be adjusted laterally with respect to the fixture to achieve a force-applying configuration of 0 mm, 1 mm, 2 mm, and 3 mm from the center. A vector network analyzer (VNA) was connected to two pins of the QCR via a BNC adapter to accurately measure the resonant characteristics and electrical parameters of the resonator. The relevant parameters measured by the VNA are obtained in real-time with specialized software in the computer. Throughout the experimental process, due to the small size of the strip QCR, the placement of the QCR and weights needs to be precisely adjusted and controlled to ensure the accuracy of the experimental results in order to achieve the corresponding force-applying position as accurately as possible. As shown in Figure 9, when the QCR was held in the experimental setup by the fixtures, the resonant frequency and electrical parameters of it changed due to the mechanical stress of the fixture. Consequently, for the sake of convenience, the parameters of the QCRs measured when the applied force was 0 N were the parameters of the QCR at the corresponding force-applying position when it was fixed by the fixture without adding weights.

In this experiment, three sets of AT-cut strip, 6 MHz QCRs with different rotation angles were purchased from Chengdu Kingbri Frequency Technology Co., Ltd. (Chengdu, China) to use, labeled QCR-1, QCR-2, and QCR-3, respectively, as shown in Figure 10. The rotation angle of QCR-1 was 30°, that of QCR-2 was 40°, and that of QCR-3 was 50°. The length of the quartz crystal plate was 9 mm, the width was 2.5 mm, and the thickness was 0.277 mm. The length and width of the silver electrodes were 7 mm and 2 mm, respectively.

## 5. Results and Discussion

To ensure the accuracy of our experimental results, we measured the frequency responses of three QCRs at different magnitudes of force and at different force-applying distances from the center. We performed these measurements over a force range of 0 N to 3 N, with force increments of 0.5 N, and we repeated them three times. By these repetitive measurements, we were able to obtain reliable and consistent results for the force sensitivity of the QCRs. As shown in Figure 11, a least squares fit of the frequency shift was performed to obtain a linear plot of the force-frequency characteristics. Frequency shifts of all QCRs showed an excellent linear increase to the applied force, with almost all regression coefficients (*R*^2^) around 0.98. The experimental error analysis graphs in Figure 11 indicate a small standard deviation of the frequency shift, suggesting high stability of the experimental environment and system. The small range of data variability further confirms the reliability of the results.

When the distance was 0 mm, i.e., the force was applied in the *X*-axis direction of the QCRs, the QCRs had the highest force sensitivity, and as distance increased, the force sensitivity decreased. When the force was applied at the edge of the QCRs, i.e., at a distance of 3 mm, the magnitude of the force did not affect the resonant frequencies. The reason for this phenomenon may be that the stresses and strains inside the QCR were different when the force was applied at different locations in the QCR. The stress distribution of the QCR was symmetrical when the force was applied near the *x*-axis and asymmetrical when the force was applied near the edge of the QCR. Additionally, as the force-applying position gradually shifted away from the *X*-axis, the stress variation in the center of the electrode decreased, which in turn led to a decrease in frequency shift. As a result, the response of the QCR to a force applied at the edge position was very small. In conclusion, the force sensitivity of the QCR varied with the location of the force-applying positions. As shown in Table 1, the force sensitivity of QCR-1 was higher than that of QCR-2, and the sensitivity of QCR-2 was higher than that of QCR-3 at the same force-applying position. This observed trend aligned with the results of our simulations, as shown in Figure 7.

The conductance diagram of QCR-3 at different force-applying positions is shown in Figure 12, and the conductance diagrams of the other two sets of QCRs are similar. Its peak conductance and center frequency decreased with the increase in applied force, and the half-bandwidth was generally broadened with the increase in force. As shown in Figure 12a, the peak conductance decreased fastest with increasing applied force. Conversely, in Figure 12d, all curves were almost coincident, indicating that when the force was applied near the edge of the QCRs, the half bandwidth and peak conductance of the QCRs were almost constant. This result might be due to the fact that the edge of the QCR was very little affected by force, so the conductance did not change significantly as the force increased.

The Q value is an electrical parameter that describes the stability of a QCR. It refers to the ratio of energy stored and energy lost by the QCR without dissipation and is an important parameter to measure the energy storage and dissipation capability of a vibration system. It is an important parameter to measure the energy storage and dissipation capability of a vibration system. Generally speaking, the higher the Q value of a QCR, the stronger its oscillation capability, i.e., the smaller the variation in output frequency and the higher the stability [36].

Figure 13 shows the Q value for all QCRs with different distances and force magnitudes. The results showed that the Q value decreased with the increase in the force magnitude. When the force was applied along the *X*-axis, i.e., the distance was 0 mm, the Q value decreased fastest, and with the increase in distance, the Q value decreased more slowly. The reason for this phenomenon is that the QCR was subjected to AC voltage during operation to produce thickness shear vibration. As the applied force increased, the amplitude of the vibration of the QCR decreased, and the friction loss of the QCR increased, resulting in a decrease in the stability of the QCR and a decrease in the Q value.

The Q value for each group of QCR remained high (>1000) over the entire force loading range of 0 N–3 N, which was still within the acceptable range and had good promise and value for practical applications.

## 6. Conclusions

In this paper, we investigated the force-frequency characteristics of the AT-cut strip QCR by experimental and finite element analysis methods. Firstly, we simulated the internal stress distribution of a strip QCR with a resonant frequency of 6 M when subjected to a pair of opposing forces in the *X*-axis direction using the finite element analysis software COMSOL Multiphysics. Subsequently, the frequency shift of the QCR under different rotation angles and different force-applying positions was simulated, and we found that the frequency shift of the QCR was maximum at a rotation angle of about 20° and almost zero around 60° when the force was applied in the *x*-axis direction. We further analyzed the strains of the QCR under this condition. Finally, based on the simulations, three sets of AT-cut strip QCRs were tested, and their resonant frequency shifts, conductance, and Q value were measured for four different force-applying positions. The experimental results showed that the frequency shift of the QCRs was proportional to the magnitude of the force, with good force-frequency characteristics. The force sensitivity of the QCR with a rotation angle of 30° was found to be the highest, followed by a rotation angle of 40° and the lowest with a rotation angle of 50°. Frequency shifts and conductance were positively correlated with the magnitude of the force and negatively correlated with the force-applying distance. In contrast, stability was negatively correlated with force and positively correlated with force-applying distance. When the force was applied along the *X*-axis, the frequency shift was maximal, with the fastest decrease in peak conductivity and stability. However, applying the force at the edge of the QCRs had essentially no effect on the frequency shifts, conductance, and stability of the QCRs. These results are important for understanding how the resonant characteristics of strip QCRs change under different force-applying positions.

## Figures and Tables

**Figure 1 sensors-23-02996-f001:**
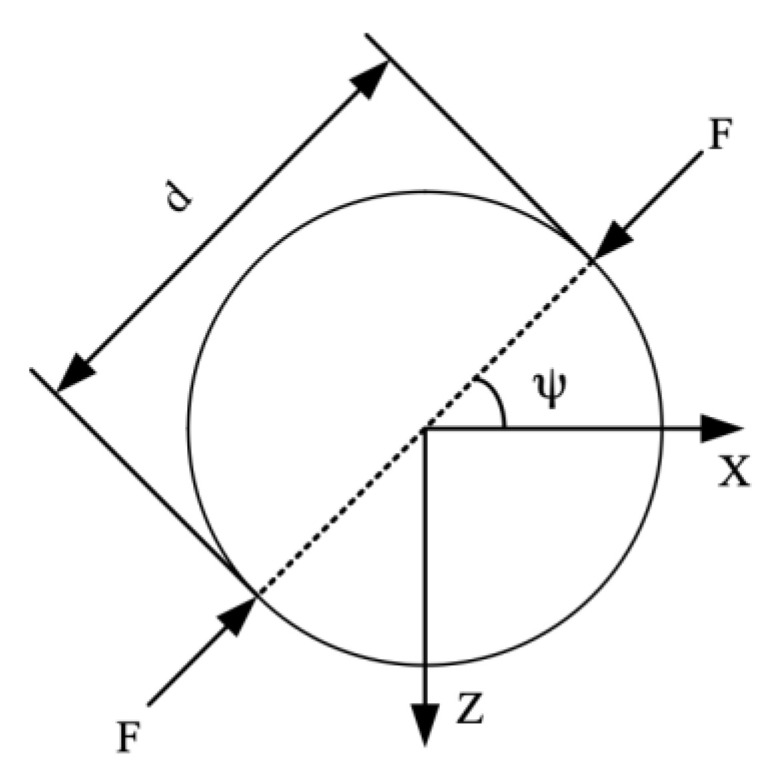
Disc-type quartz crystal subjected to two opposed radial forces.

**Figure 2 sensors-23-02996-f002:**
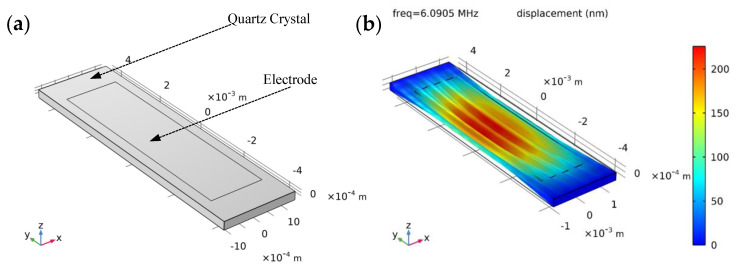
(**a**) Finite element model of AT-cut strip QCR; (**b**) Displacement of the QCR resonating in its resonant frequency of 6.0905 MHz.

**Figure 6 sensors-23-02996-f006:**
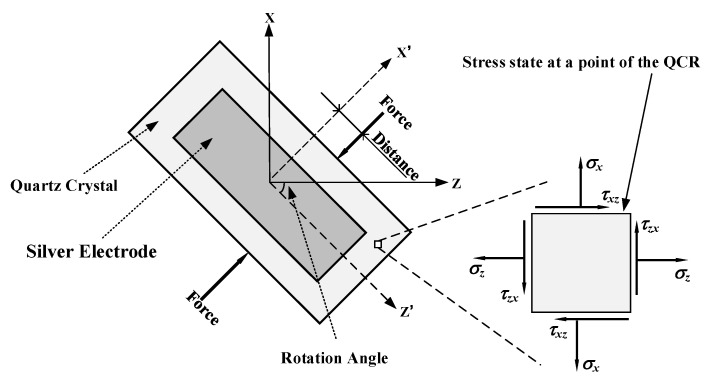
The force-applying configurations of the strip QCR.

**Figure 7 sensors-23-02996-f007:**
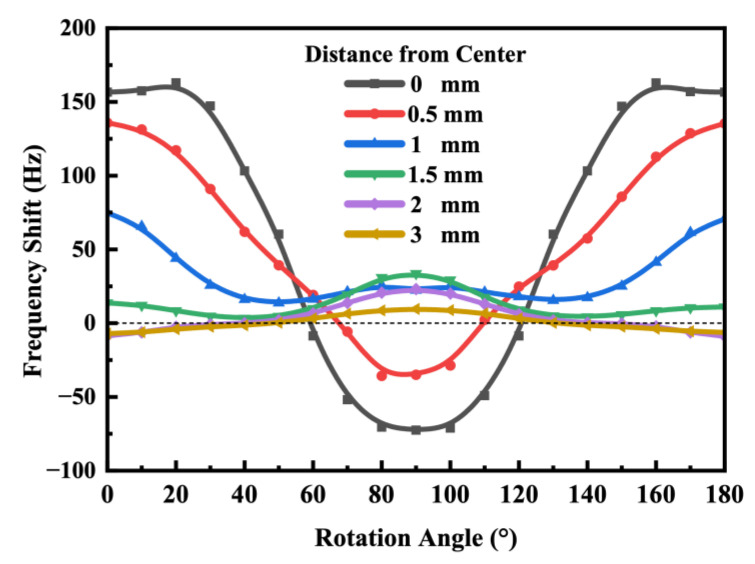
Relationship between frequency shift and rotation angle of AT-cut strip QCR under different force-applying positions.

**Figure 8 sensors-23-02996-f008:**
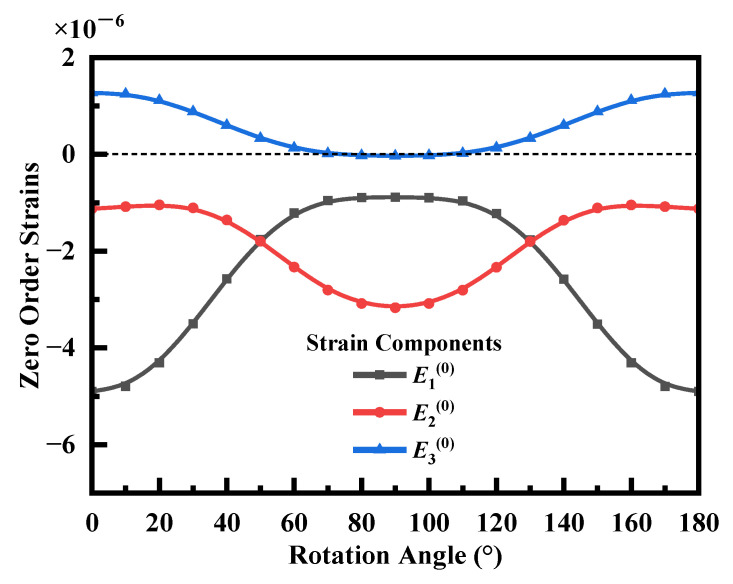
Relationship between zero-order strains and the rotation angles.

**Figure 9 sensors-23-02996-f009:**
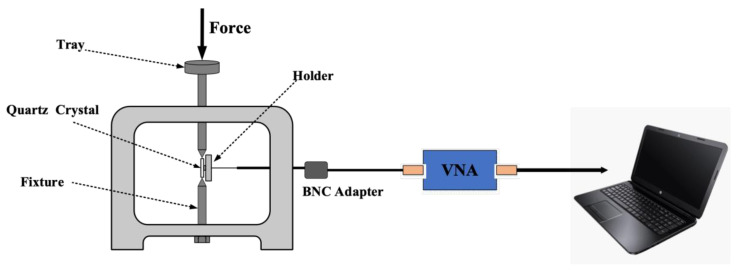
Schematic diagram of the loading fixture for the force-frequency experiment.

**Figure 10 sensors-23-02996-f010:**
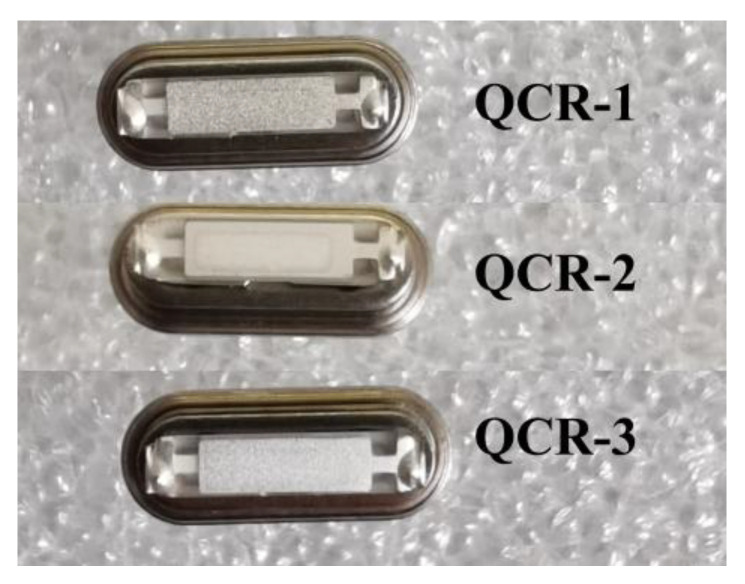
Photos of QCRs.

**Figure 11 sensors-23-02996-f011:**
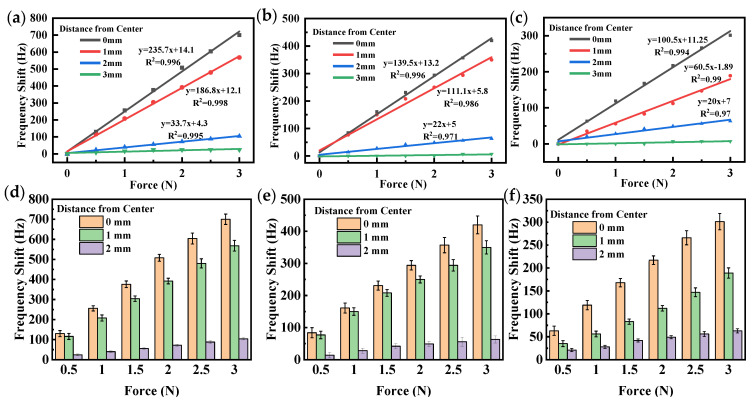
Fitting line graph of frequency shift vs. applied force for (**a**) QCR-1; (**b**) QCR-2; (**c**) QCR-3, and experimental error analysis for (**d**) QCR-1; (**e**) QCR-2; (**f**) QCR-3.

**Figure 12 sensors-23-02996-f012:**
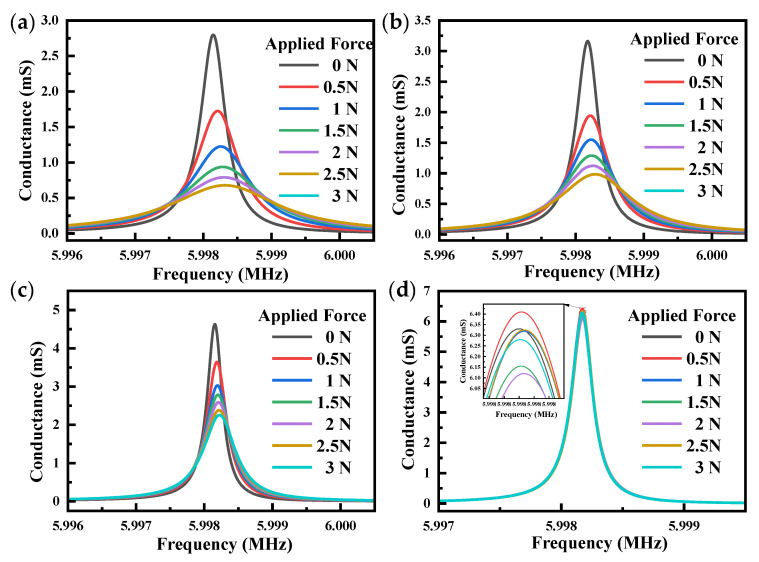
Conductance of QCR-3. (**a**) distance = 0 mm; (**b**) distance = 1 mm; (**c**) distance = 2 mm; (**d**) distance = 3 mm.

**Figure 13 sensors-23-02996-f013:**
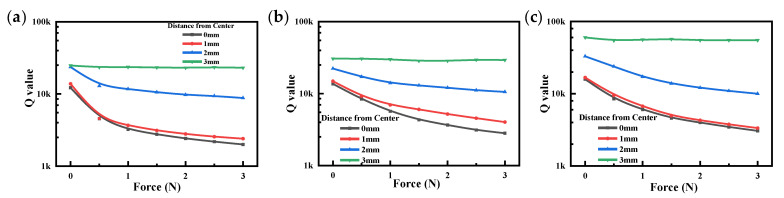
Q value vs. applied force. (**a**) QCR-1; (**b**) QCR-2; (**c**) QCR-3.

**Table 1 sensors-23-02996-t001:** Force sensitivity of AT-cut strip QCRs under different force-applying positions.

	Distance = 0 mm	Distance = 1 mm	Distance = 2 mm
QCR-1	235.7 Hz/N	186.8 Hz/N	33.7 Hz/N
QCR-2	139.5 Hz/N	111.1 Hz/N	22 Hz/N
QCR-3	100.5 Hz/N	60.5 Hz/N	20 Hz/N

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
