# Peer review of "Study of Force-Frequency Characteristics in AT-Cut Strip Quartz Crystal Resonators with Different Rotation Angles"

_sensors, 2023, doi:10.3390/s23062996_

Round 1

Reviewer 1 Report

Yang and coworkers report Force-Frequency Characteristics in AT-cut Strip 2Quartz Crystal Resonators with Different Rotation Angles. The data are solid and seem to be convincing, the paper is well organized. The paper could be published after minor considerations.

1. Add the detailed discussions of Figures 3, 4 and 5.

2. Pls explain that the mechanism of frequency shift of the QCR, why  maximum and zero are at a rotation angle of about 20 and almost zero around 60 degree, respectively.

3. Quartz has piezoelectricity. Will the anisotropic piezoelectricity of Quartz crystal affect the tested performance.

4. What is the full name of AT-cut? When other abbreviations first appear, the full name should be defined?

5. As for Figure 9, is there any discussion on error analysis for the test?

Reviewer 2 Report

This paper provide results on the frequency change with the concentrated force applied in the edge of a quartz crystal resonator.  The objective of this study is to guide the design of a force sensor.  Such calculations are needed in producrt development.

There are some issues to be discussed further:

1) How the force is applied to the crystal blank?  Is it applied to a holder as a buffer?  Details are needed.

2) If the crystal is rotated, are the material constants are also modified?  How?

3) Sensors are usually made with circular resonators for obivious reasons.

4) If the forced is exterted in a rectangular plate, the analysis can be done with plate theory (Ji Wang, J.-D. Yu, and Yook-Kong Yong: On the correction of the higher-order Mindlin plate theory, International Journal of Applied Electromagnetics and Mechanics, 22: 83-96, 2005; Ji Wang and E. Momosaki: Piezoelectrically forced vibrations of AT-cut quartz strip resonators, J. Applied Physics, 81(4): 1868-76, 1997. [doi:10.1063/1.364042]).

5) Reference 19 spelling error (Guanping).

The paper has to be revised to address the above questions.

Reviewer 3 Report

The Authors study the force-frequency characteristics of AT-cut strip quartz crystal resonators, both numerically (using COMSOL Multiphysics) and experimentally, as a function of the force-application position and rotation angle.
The results are well described and interesting; therefore, I suggest its publication with some minor changes.
1) line 27: "of its compact" -> "due to its compact"
2) line 38: "present" -> "presented"
3) line 74: "serval" -> "several"
4) line 147: "position that" -> "position such that"
5) in Section 4 some details should be added on the electrical quantities applied during the measurement
6) line 201: "As shown in Figure 10d, all" -> "In Figure 10d, all"
7) lines 160, 230: please clarify what you mean by "stability" (if the
constancy of Q).

Reviewer 4 Report

Discussion can still be improved, bringing more differences and innovations when compared to other works and state-of-the-art.

The instrumentation, data analysis, statistics (paragraph 1, 2) are insufficiently described or represented.

Figure 1 is too large.

The conclusions could be more detailed: maybe a comparison between the experimental results and the simulation results

Reviewer 5 Report

Study of Force-Frequency Characteristics in AT-cut Strip

Quartz Crystal Resonators with Different Rotation Angles

In this work author studied the force – frequency characteristic in AT-cut Strip Quartz Crystal Resonators with Different Rotation Angles through both software and experimental analysis.

The paper is well-written and easy to comprehend. However, I have a few queries which need to be addressed by the authors:

1.    In introduction section the literatures 1-12 are need to be discuss elaborately.

2.    At the end of introduction section, the summary of literature review, research gap is not clear.

3.    The organization of proposed research work is need to add the end of introduction section for getting more clear view about the proposed work.

4.    In outcome of conducting experimental and simulation analysis need to mention in result and analysis section for more visibility.

Round 2

Reviewer 2 Report

This reviewer has noted that the paper has been revised by the authors extensively.  Earlier questions from the reviewer have been addressed.  The answers are satisfactory.

The paper is recommended to publish with some minor changes:

1) The effect of force in X-direction should be dsicussed because wave is propagating in the X-direction.  Will it be more sensitive?  The effect can be more smooth.

2) Check the term S_ij in Eq. (4).  Generally we call them elastic constants and compliance constants.  Make sure the popular terms are used.  Flexibility is NOT popular term.  Check textrbooks.

3) In case a paper has a journal version, the journal paper should be cited, like Ref 25.

4) The word "zero-order" should be "zeroth-order".

5) Some refs like 25, 27, 28 should use the journal style.  It is: author, paper tile, In Proceedings of ..... Pay attention to the journal style.
